# A Facile Synthesis Process and Evaluations of α-Calcium Sulfate Hemihydrate for Bone Substitute

**DOI:** 10.3390/ma13143099

**Published:** 2020-07-11

**Authors:** Nhi Thao Ngoc Le, Ngoc Thuy Trang Le, Quang Lam Nguyen, Truc Le-Buu Pham, Minh-Tri Nguyen-Le, Dai Hai Nguyen

**Affiliations:** 1Institute of Applied Materials Science, Vietnam Academy of Science and Technology, Ho Chi Minh 70000, Vietnam; lethaongocnhi@gmail.com (N.T.N.L.); atomnguyen2113@gmail.com (Q.L.N.); trihankiner215@yahoo.com (M.-T.N.-L.); 2Institute of Research and Development, Duy Tan University, Danang 550000, Vietnam; lenthuytrang4@duytan.edu.vn; 3Biotechnology Center of Ho Chi Minh City, Ho Chi Minh 70000, Vietnam; buutruc@hcmbiotech.com.vn; 4Graduate University of Science and Technology, Vietnam Academy of Science and Technology, Hanoi 100000, Vietnam

**Keywords:** α-calcium sulfate hemihydrate (CaSO_4_·0.5H_2_O), autoclave method, surgical grade, in vitro biocompatibility, cytotoxicity

## Abstract

Alpha-calcium sulfate hemihydrate (α-HH) has been used effectively in grafting through its desired features to support bone regeneration. In recent years, many synthetic methods have been proposed. Among them, the autoclave method for manufacturing α-HH is best suited for cost-savings due to its simple operation and limited use of additives. Despite these advantages, the synthesis of surgical grade products without the use of any additives has not yet been clearly discussed. In this study, surgical grade α-HH was successfully produced from calcium sulfate dihydrate (DH) using the autoclave method at an elevated temperature and pressure. The synthesized powder had a high purity of about 98.62% α-HH with a prismatic morphology (20.96 ± 8.83 µm in length and 1.30 ± 0.71 µm in diameter). The screening tests, in simulated body fluid (SBF) solution, for the product properties showed no bioactivity, and fast degradation accompanied by a slight decrease in pH. The lactate dehydrogenase (LDH) assay showed good biocompatibility of the material, however, its potential for cytotoxicity was also observed in NIH 3T3 cells. Briefly, despite some unfavorable properties, the autoclave-synthesized α-HH is a promising bone graft substitute that can be applied in orthopedic and maxillofacial surgeries.

## 1. Introduction

Bone regeneration, or fracture healing, is a fascinating process in which bone is able to self-regenerate and reach its initial function without leaving any types of scar tissue [1]. Despite its potential for robust healing, there are conditions in which bone fails to heal leading to delayed or nonunions as systemic bone loss (i.e., osteoporosis and osteopenia) and large bone defects due to trauma or cancer [2]. This is a substantial medical issue because treating such conditions remains a challenging clinical scenario for the treating physician, and especially a growing financial burden for patients. Although many efforts have been made for bone reconstruction, they all have specific indications and limitations. Established methods are bone transport distraction osteogenesis, or bone grafting, including autografts, allografts, xenografts, and synthetic substitutes [3,4]. Autografts are the most effective bone graft materials because they contain live cellular component. However, the operative time of autografts and the donor site morbidity increase due to the harvesting of a graft from the patient’s own body [5]. Thus, the introduction of alternative sources is necessary. In particular, the use of synthetic bone substitutes is advantageous because they help avoid the problems of finding suitable bone from within the body (autografts) and the infective risks associated with the use of human cadaver materials (allografts) or other animal tissue (xenografts) [5,6]. Calcium sulfate (CS) is the simplest bone substitute and has been widely recognized as a well-tolerated and readily available material with the most affordable price and prolonged history of clinical use for grafting [6,7]. Calcium sulfate benefits from a crystalline structure and biodegradation to support space for cell growth and increases extracellular calcium ions [1,4]. Thus, it promotes osteoconduction, regeneration, and hemostasis when placed in surgical/periodontal bone defects, where it resorbs the blood proteins to start the intrinsic coagulation pathway, and then is fully absorbed and leaves behind a calcium phosphate lattice [8,9]. Furthermore, CS seems to stimulate angiogenesis, i.e., the formation of new blood vessels, leading to improved osteogenic processes [6]. Calcium sulfate exists in the following three crystalline phases: dihydrate (DH, CaSO_4_·2H_2_O), hemihydrate (HH, CaSO_4_·0.5H_2_O) and anhydrite (AH, CaSO_4_). In terms of hemihydrate, there are two distinction structure, i.e., α and β [10,11]. Among all the polymorphs of CS, so far, α-calcium sulfate hemihydrate (α-HH) is studied the most for medical applications because of its proper mechanical properties [7,12]. The manipulation of α-HH as bone grafts has long been known and used, with proven efficacy in patients [13,14,15]. It is easily molded and sterilizable with gamma radiation, before use in surgical procedure. Accordingly, using α-HH could also cut down on handling costs, giving patients more opportunities for proper treatment.

In nature, calcium sulfate hemihydrate is rare but dihydrate is the most abundant sulfate mineral in the Earth’s crust [12]. Thus, it is profitable to synthesize α-HH from dihydrate in order to reduce the therapeutic expenditure for patients in bone grafting. Various methods for producing α-HH have been reported, namely: (i) autoclave method, (ii) oxidation process, (iii) salt solution method, (iv) electrochemical deposition, (v) microemulsion method, and (vi) microwave–assisted route [16,17,18,19,20]. The autoclave method is the only method, so far, that has been employed to produce α-HH from dihydrate (DH) for large quantity synthesis due to its simple operation [11,17]. In addition, for other methods, additives (salts, acids, organic solvents, and surfactants, etc.) are essential to meet the conditions for the phase transition or to act as modifiers for homogeneous crystal morphology. Consequently, these chemicals can be integrated into α-HH crystals during the synthetic process. When applied, the human body can produce unwanted immune reactions with such impurities, and thus new FDA approval criteria are required [21]. The first commercial processes for α-HH production were used in building construction in Germany, Japan, and America [16,22]. As a traditional method used in industry, the production of α-HH by autoclaving has long been known [17], however, the output of high purity for medical application, especially when not adding any additives in processing, has hardly been studied.

With the focus of reducing expenses and associated clinical risks with bone treatments, the present study provides a one-step synthetic process of α-HH from dihydrate. The synthesized α-HH was obtained at the surgical grade and minimized potential risks from additives. For additional information on the biological properties, preliminary assessment in simulated body fluid (SBF) was also carried out. A cytotoxicity assay was used to investigate the potential toxic effects of α-HH, particularly in the form of calcium sulfate particles, which can exist clinically in patients.

## 2. Materials and Methods 

### 2.1. Dehydration Process into Hemihydrate in Production

Calcium sulfate dihydrate powder (98%; Acros Organics–Thermo Fisher Scientific Inc., Morris Plains, NJ, USA) was used as a precursor in this study. Firstly, 5 g CaSO_4_·2H_2_O was added into a glass laboratory bottle containing 200 mL of distilled water and stirred at a speed of 600 rpm for 15 min. Then, it was brought to synthetic conditions at 140 °C, and pressure of 2.7 MPa for 4 h in an autoclave. After the reaction ended, the suspension was immediately filtered and rinsed five times with boiling distilled water. Finally, an acetone washing step was employed to remove water followed by drying at 55 °C, for 16 h, to remove acetone. The synthesized powder was preserved in a dark glass container at room temperature for study.

### 2.2. Hydration Process into Dihydrate in Usage

The synthesized powder was mixed with deionized water with a liquid to powder (L/P) ratio of 0.8 mL/g. The mixture was stirred to form homogeneous paste, then, injected into polystyrene molds (14 mm diameter × 4 mm thickness) and stored in 65% humidity at 37 °C for 24 h. The formed disc-shaped samples corresponding to the usable form of α-HH, also known as calcium sulfate cements, were characterized and evaluated for their bioproperties.

### 2.3. In Vitro Preliminary Assessment in SBF

The simulated body fluid (SBF) is an inorganic solution with ion concentrations similar to human blood plasma. SBF solution has been the best solution, so far, to check the in vitro bone bioactivity through the apatite-forming ability of implant surfaces intended to come into direct bone contact [23,24]. Apatite is chemically similar to the main inorganic component of mammalian bone tissue. It is one of the few materials that is classified as bioactive material, meaning that it supports bone ingrowth and osseointegration when being implanted into a living body [25]. Increased biological activity leads to apatite forming on the material surface in a shorter time.

The Tris-HCl-buffered SBF solution of 27 mM HCO^3−^ (namely Tris-SBF-27), used in this study, was formulated based on Kukobo’s specification [24] with modifications from A. Cuneyt Tas et al. [26] to better match with ion concentrations of human plasma. All chemical reagents were analytical reagents (AR) and were used directly without any purification. The materials were monitored for different immersion periods in the solution (1, 3, 5, 7, and 10 days). The ratio of total surface area to Tris–SBF-27 solution volume was 10 mm^2^ mL^−1^ at pH 7.4 and 37 °C for each disk-shaped sample. The solution was refreshed every 24 h to ensure constant initial chemical composition [27]. The pH value of the solution was measured daily for a group of samples without refreshing. After the preselected soaking time, the samples were washed gently with deionized water to remove the SBF solution followed by air-drying in a desiccator. Then, the dry mass was weighed to calculate the change before and after the immersion in the SBF solution. The degradation was calculated in terms of percentage by the weight loss per its initial weight.

### 2.4. Material Characterization

The morphology images of samples were taken with a scanning electron microscopy (SEM) (S-4800; Hitachi Ltd., Tokyo, Japan) and dimensioned using ImageJ software (ImageJ; version 1.52a, National Institutes of Health, Bethesda, MD, USA). Crystallinity analysis and phase identification were carried out by infrared spectrum (FTIR) (Frontier FTIR/NIR spectrometer; PerkinElmer Inc., Waltham, MA, USA) in the 4000–400 cm^−1^ wavenumber range using the KBr pellet method, and X-ray diffraction (XRD) (D8 Advance-Bruker X-Ray Diffractometer; Bruker AXS GmbH, Karlsruhe, Germany) with Cu Kα radiation (λ = 1.54178Å), at a scanning rate of 5°/min in the 2θ range from 5° to 70°. Thermogravimetric analysis (TGA) was performed to determine the purity of the phases using a thermal analysis system (TGA-DSC 3+; Mettler Toledo Inc., Columbus, OH, USA) from 0–300 °C at a heating rate of 10 °C /min in the air.

### 2.5. Lactate Dehydrogenase (LDH) Cytotoxicity Assay

In the present study, LDH cytotoxicity assay was performed in direct contact between cells and CS particles for short-term exposure evaluation. The mouse embryonic fibroblast cell line (NIH 3T3; Biotechnology Center of Ho Chi Minh City, Ho Chi Minh City, Vietnam) was cultured in Dulbecco’s modified Eagle medium (DMEM) (Gibco–Thermo Fisher Scientific Inc., Waltham, MA, USA). The synthesized powder was weighed at 10, 25, 50, 100, and 200 mg. These samples were, then, sterilized with ultraviolet radiation for 2 h. The sterilized samples were soaked in 1 mL cell culture medium for 1 h of sonication to obtain sample solutions. The NIH 3T3 cell suspensions with 1 × 10^5^ cells/well per 100 µL were seeded to each well and incubated under cell culture conditions for 24 h to form a semi-confluent monolayer. Subsequently, cell culture mediums were replaced with 100 µL of sample solutions (n = 7). The cytotoxicity after incubation for 24 h was determined using the Cytotoxicity LDH Assay Kit-WST (Dojindo Molecular Technologies Inc., Kumamoto, Japan) according to the manufacturer’s instructions. Following the incubation for 4 h, the formazan crystals were formed.

The membrane integrity was visualized based on the cleavage of a tetrazolium salt (WST) to red color formazan by the released intracellular lactate dehydrogenase (LDH) in living cells. The cell viability increased in proportion to the amount of formazan formed, which was recorded by the optical density (OD) value at 450 nm using a microplate reader (VersaMax™ Microplate Reader; Molecular Devices LLC, Sunnyvale, CA, USA). The percentage viability was calculated from the OD value using the following equation:$$\mathrm{Viability}\;\left(\%\right)=\frac{{\mathrm{OD}}_{s}-{\;\mathrm{OD}}_{b}}{{\mathrm{OD}}_{n}-{\mathrm{OD}}_{b}}\times 100$$
where ODs is the mean value of the measured OD of the tested samples; ODn is the mean value of the measured OD of the negative control samples; and ODb is the mean value of the measured OD of the blank samples. The sample is considered to have acute cytotoxic potential when the viability value declines less than 70%.

### 2.6. Statistical Analysis

The datasets were processed using Microsoft Excel 2019 software and were expressed as mean ± standard deviation. The experimental mean values were compared by one-way analysis of variance (ANOVA) using Minitab software (Minitab^®^; version 16.0, Minitab LLC, State College, PA, USA) with the Tukey’s test for comparison between groups. In all evaluations, *p* < 0.05 was considered to be statistically significant.

## 3. Results and Discussion

### 3.1. Characterization of the Synthesized Powder

The Fourier transform infrared (FTIR) (Figure 1) shows absorbances of crystalline components in the synthesized powder. The principal peaks corresponding to hemihydrate represented at 660 cm^−1^ for bending vibrations of SO_4_^2−^ ion, 3560 and 3610 cm^−1^ for O–H valence (stretching), and 1620 cm^−1^ for bending vibrations of a single type of water in molecular structure [28,29]. The other peaks denoted vibrations of sulfate ion including 600 cm^−1^ (bending), 1008 cm^−1^ (stretching), 1096 cm^−1^ (stretching), 1115 cm^−1^ (stretching), and 1152 cm^−1^ (stretching). There was no presence of foreign functional groups in crystals which showed that acetone in the washing step was effectively removed.

The XRD pattern (Figure 2) further confirms the formation of calcium sulfate phases by their typical peaks. Most peaks were indexed corresponding to standard HH (ICDD 41-0224). Three peaks corresponding to standard DH (ICDD 33-0311) were also detected at a low signal. The results revealed that HH was almost the only mineral in the synthesized powder, but there were still traces of DH. The XRD was still unable to verify whether the hemihydrate belonged to α-form or β-form. Nevertheless, Surajit M. et al. pointed out from the standard patterns of CS phases that β-HH would have broad and low intensity diffraction peaks [30]. Additionally, the peak of α-HH at (204) plane would be remarkably lower than (400) plane. The present synthesis using an autoclave resulted in the formation of highly crystalline crystals as opposed to β-HH and crystal planes compatible with α-HH. Consequently, the synthesized calcium sulfate hemihydrate was expected to be in α-form.

Figure 3 shows the morphological changes before and after dehydration treatment. From the micrographs, the dihydrate precursor with varied shapes (Figure 3a) was converted to hemihydrate crystals which have homogeneous morphology of prismatic shape with sharp edges (Figure 3b). It is widely accepted that α-HH crystals are characterized by well-formed transparent idiomorphic crystals with sharp crystal edges, whereas β-HH consists of flaky particles made up with small crystals [10,15,31]. Therefore, combined with the XRD results in Figure 2, it can be strongly concluded that the as-synthesized hemihydrate was ascribed to α-form. Size measurements gave an average crystal length of 20.96 ± 8.83 µm and diameter of 1.30 ± 0.71 µm.

Figure 4 shows the weight loss (%) of the synthesized product measured by TGA. Theoretically, the crystal water content of HH and DH is 6.2% and 20.9%, respectively. Starting from an ambient temperature, the curve showed that the powder gradually lost a small amount of mass through physisorbed water. The rapid mass loss due to the dehydration (at around 100 °C) of sample was recorded at 6.4026%, which is greater than the percentage of water by theory of HH [32,33]. Therefore, the phase analysis in TGA was compatible with the XRD results in Figure 2, indicating that HH and DH were both included. Calculating from the recorded weight loss, the purity of α-HH was estimated to be 98.62%. Consequently, with the chemical minimization in production, the α-HH has sufficiently reached surgical grade in the FDA standard for the purity (CaSO_4_ ≥ 98 mass%) to be applied as bone materials [21].

### 3.2. In Vitro Studies of Cement Samples in SBF

#### 3.2.1. Bioactivity Assessment

Calcium sulfate hemihydrate (α-HH) powder, when mixed with water, readily hydrates into calcium sulfate dihydrate (DH). This reaction took place to convert the as-synthesized α-HH powder into a form that can be used in medical treatment, namely calcium sulfate cement [10,11]. Figure 5 exhibits FTIR spectra and XRD patterns of the cements before and after 1, 3, 5, 7, and 10 days of immersion in the Tris–SBF-27 solution.

In FTIR analysis (Figure 5a), the peak at 670 cm^−1^ was typical for bending vibrations of SO_4_^2−^ ion in dihydrate (DH). The bands at 3610 and 3560 cm^−1^ in HH were lowered to 3545 and 3402 cm^−1^ in DH, respectively, indicating the addition of hydrogen bonds in O–H stretching. Two bending vibrations at 1686 and 1622 cm^−1^ indicated the presence of two types of water molecules in DH crystals. The other vibrations of DH are shown at 603 cm^−1^ (SO_4_^2−^ bending), 1005 cm^−1^ (SO_4_^2−^ stretching), 1140 cm^−1^ (SO_4_^2−^ stretching), and 3246 cm^−1^ (H_2_O stretching) [28,34]. There were no typical peaks of apatite that appeared, which would be around 1455–1420 cm^−1^ for CO_3_^2−^ stretching vibrations, 1030 cm^−1^ for PO_4_^3−^ stretching vibrations, and 870 cm^−1^ for bending vibrations of CO_3_^2−^ substituting for PO_4_^3−^ in apatite lattices [26,31]. The XRD patterns (Figure 5b) revealed all characteristic peaks of DH and remained unchanged after the immersion. There were no impurity peaks as compared with the nonimmersion samples that appeared in both FTIR and XRD. These results pointed out that α-HH was completely hydrated into DH and did not convert or create any other substance during the immersion period.

The SEM micrographs of the cement samples after the immersion test in the SBF (Figure 6) display porous structures consisting of interlocking crystals in the form of large and irregular particles. When samples were immersed in the SBF, the dissolution of CS occurred leading to a gradual change in shape of the crystals from rough plates into needles with smooth surfaces as a function of the immersion time. These morphologies were also described by N. B. Singh and B. Middendorf [10], Jalota S. et al. [26], and Chen Z. et al. [32]. XRD, FTIR, and SEM provided evidence proving that the forming crystals are of dihydrate (DH). Many in vitro and in vivo studies have indirectly or directly revealed that CS has always failed to form bonds with bone tissue [13,33,35]. This is also supported by the results in this study that no microspheric apatite aggregated on the crystal surface of cement samples after 10 days immersed in Tris–SBF-27 solution. The bioactivity are assessed through the newly formed apatite layer, which helps living bone tissue cling to the material surface [24,31,35]. Therefore, it can be deduced from these results that CS lacks bioactivity for better bone regeneration in defects. An advantage is that the by-products of CS degradation did not consist of foreign substances making CS harmless if being degraded in the human body.

The nature of calcium sulfate is less beneficial for bone regeneration due to the lack of bioactivity gained from apatite layers as mentioned above. However, it has the advantage of being used as an inorganic binder to fill defects for bone regeneration, thanks to the setting process as happened to form the cements [11,15]. It is easily molded into the bone defects and acts as a barrier to prevent excessive invasion of soft tissue [4]. Meanwhile, CS still provides sufficient space for osteoconduction, neovascularization, and osteogenesis owning to its porous structure [4,36]. Furthermore, when CS cement is placed in bone defects, it begins to dissolve and releases calcium (Ca^2+^) and sulfate (SO_4_^2−^) ions [1,37]. The ions are not sufficient to induce apatite nucleation but can act as a reservoir of calcium ions for bone reconstruction in patients [37]. Moreover, local increases in calcium ion concentration can also induce osteogenesis by establishing interaction between the osteoclasts (OCs) and osteoblasts (OBs) with Ca-sensing receptors (CaR) [1,7].

#### 3.2.2. Degradation and pH Alteration of Cement Samples

The weight calculation for degradation rate and pH measurement of CS cements are presented in Figure 7. It was found that the weight loss of the cements increased over time. The degradation process occurred vigorously when CS first came into contact with the solution, and then, tended to progress gently and reached the rate of 59.89% at the end of the testing period. The pH value of SBF solution dropped sharply within three days (from 7.4 to 7.0), and was maintained after soaking for 10 days.

As a disadvantage feature of CS, the recorded weight loss was compatible with the fast degradation of CS reported in previous literatures [33,35]. Apparently due to the fast degradation, the resulting calcium-rich fluid led to a decrease in pH, which suggested that acidic products were produced in the solution. In the human body, the biological processes and the chemical transformations of cells are mostly carried out within a pH range of 6–8 [38]. Proteins and enzymatic rates can suffer from changes in response to modification of the pH of the surrounding fluids [39,40]. A profound alteration of pH can lead to the disruption of protein structure and loss of its function [39]. Several studies have reported that degradation products, which were released by CS, would produce a mild acidic microenvironment [31,35]. In agreement with previous studies, the pH variations of CS cements, found in this study, slightly reduced to just under neutral, therefore, they remained within the permissible range of the human body.

Degradation and pH alteration in calcium sulfate when it is being applied are both a virtue and vice. Its rapid and complete resorption can, under some circumstances, be disadvantageous due to insufficient support time for the host bone to grow into the defect area [7,26,35]. Rapid dissolution leads to slightly acidic cytotoxic microenvironments and Ca^2+^ accumulation in the surrounding fluid responsible for local inflammatory processes to take place in the early stage after implantation [40]. The incipient inflammation has three functions as follows: (1) to clean the fracture region from dead cells, (2) to initiate processes to restore the blood supply, and (3) to congregate mesenchymal stem cells [41]. It is a crucial biological process for eradication of pathogens and maintenance of tissue homeostasis. The crosstalk between inflammatory cells (leukocytes and cells of the monocyte-macrophage-osteoclast lineage) and cells related to bone healing (cells of the mesenchymal stem cell osteoblast lineage and vascular cells) is essential to the formation, repair, and remodeling in bone [42]. The onset of the inflammatory process through CS can contribute to bone regeneration but can occur excessively leading to immoderate inflammation. Permanent soft tissue alterations can result due to chronic inflammation, where active inflammation, fibrosis, and attempts at repair all occur simultaneously [40]. The α-HH, at the surgical grade, has already been used for decades in grafting as bone defects fillers [7] and 13.8–19.0% incidences of associated complications have been reported in clinical studies [13,14,40]. Still, there are no statistically significant factors that could predict its development [14,40]. The inflammation reaction is essentially self-limited and quite benign that can be observed and treated with anti-inflammatories. It is well known and this can confidently be reported to patients before surgery along with the other risks of surgery [14].

### 3.3. Cytotoxicity Tests of the Synthesized Powder

Materials cannot be used as implants if they are not biocompatible with humans. The cytotoxicity is one of the main tests that provides evidence for the biocompatibility in the context of assessing human risks for exposure scenarios with materials [25]. A cytotoxicity assay should be compatible with the physicochemical properties of the test material. Calcium sulfate is a hydrophilic substance that tends to release ionic components capable of altering intracellular enzyme activities [40]. Therefore, the LDH assay is an appropriate method to verify changes in cell membrane integrity, which is a vital mechanism of assessing cell viability/toxicity. The quantitative difference in the viability of NIH 3T3 cells treated with various CS concentrations (Figure 8) was statistically significant (*p* < 0.05).

The investigations of intracellular enzyme activities, after 24 h of acute exposure, revealed no signs of cytotoxicity at low CS concentrations (5, 10, and 25 mg/mL) with the viability over 85%. No significant difference in cell viability was observed between the group treated with 5 mg/mL as compared with the negative controls and the group treated with 10 mg/mL, as well as between the group treated with 10 mg/mL as compared with the group treated with 25 mg/mL (*p* < 0.01). It is apparent that a high cell viability rate indicated that CS manifested good biocompatibility [32,33]. However, at higher concentrations (groups treated with 50, 100, and 200 mg/mL), the synthesized powder exhibited reductions in cellular activities, suggesting its transient cytotoxicity. The main reason for this unfavorable feature has been assumed to be the rapid degradation of CS leading to an enhanced Ca^2+^ concentration [43], which was clearly shown in Figure 7. Moreover, the phagocytosis of calcium sulfate particles has also been pointed to cause death in cells [43,44].

Biocompatibility requires a risk assessment in which a cytotoxicity test must be performed to verify if calcium sulfate is capable of causing toxic responses in patients. Most materials at some concentrations produce a toxic reaction [45]. Although CS is characterized as a biocompatible material [15], cell behavior varies depending on the material characteristics and the testing system [43]. Thus, various in vitro testing methods are useful to verify if the material is potentially toxic in different ways [45,46]. CS particles induced the adverse cell response at high concentrations, however, this effect could be expected efficiently compensated in vivo. The question is, “Could patients be exposed to the source of concentrations of formed material that cause a toxic response?” In medical treatment, CS is utilized in the form of hard-setting cement, and therefore the chance of releasing a large quantity of CS particles at a time from the solid mass is relatively low. Consequently, CS has not been reported to cause substantial adverse reactions in clinical studies [13,14,40]. Nevertheless, this is only a proper level of chance with the solidified mass, and therefore we cannot exclude the possibility due to human manipulation of α-HH powder when performing surgery, especially in the event of an incomplete in situ setting process of calcium sulfate, causing the release of CS particles to reach a toxic level.

To conclude, regardless of the good biocompatibility of the synthesized powder, particulate-mediated cytotoxicity was noticed. Calcium sulfate is FDA approved and widely used because of the benefits it provides to patients. The results of cytotoxicity tests should be viewed and cautioned in the context of a complete biocompatibility test for the most accurate picture of the potential biological risks associated with the clinical use of material.

## 4. Conclusions

In this study, a facile one-step autoclave method was developed to synthesize α-HH with high purity. Using dihydrate as a raw material, without adding additives, the final product successfully reached the surgical grade. Hence, the synthesized product could potentially be used in human applications. The bioproperties of α-HH, determined in a preliminary assessment in SBF have both advantages and disadvantages, including fast degradation, pH alteration, and no bioactivity. The cytotoxicity results indicated that the synthesized powder did not influence the viability of NIH-3T3 cells, suggesting its biocompatibility. Nevertheless, at high concentrations of CS particles, a reduction in cell viability was observed. The undesirable factors were introduced that could have caused this result represent a cautious issue which needs to be considered in clinical treatment. Generally, the instability of CS cements caused by the incomplete setting process leads to the strong release of its particles which could be harmful when in direct contact with host tissues in the body. An in vivo study has to be performed for more detailed justification.

## Figures and Tables

**Figure 1 materials-13-03099-f001:**
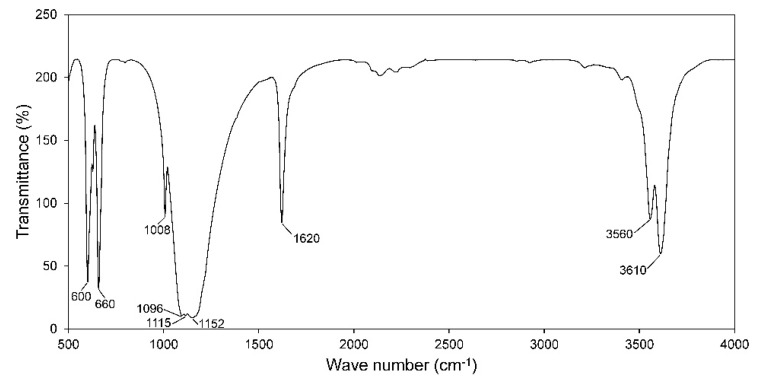
Fourier transform infrared (FTIR) spectroscopy of the synthesized powder.

**Figure 2 materials-13-03099-f002:**
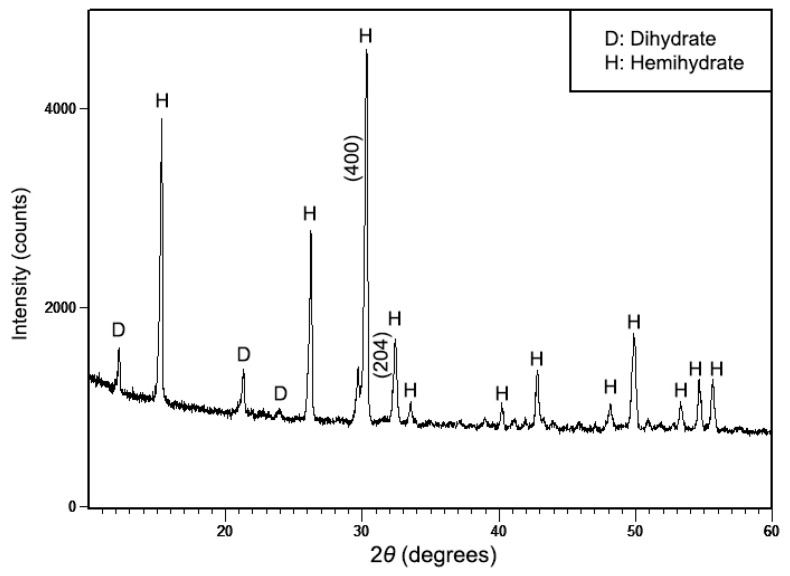
X-ray diffraction (XRD) pattern of the synthesized powder.

**Figure 3 materials-13-03099-f003:**
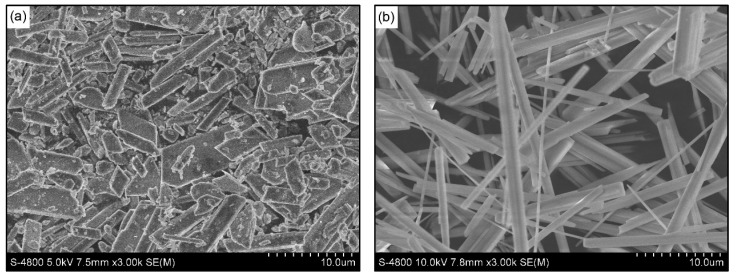
Scanning electron microscopy (SEM) micrographs of (**a**) Dihydrate (DH) precursor and (**b**) the synthesized powder.

**Figure 4 materials-13-03099-f004:**
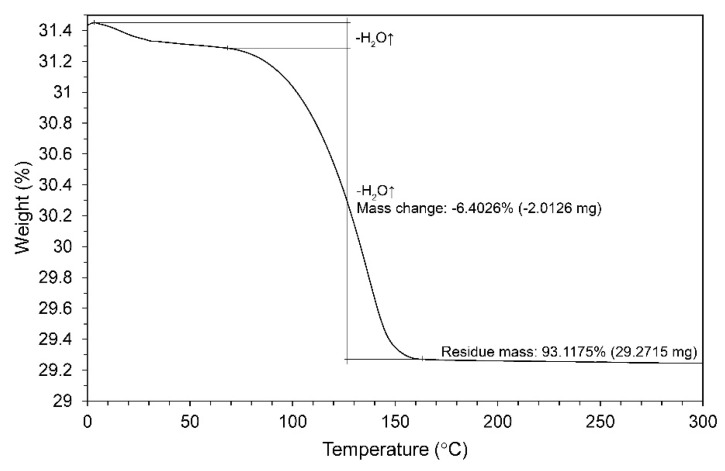
Thermogravimetry curve of the synthesized powder.

**Figure 5 materials-13-03099-f005:**
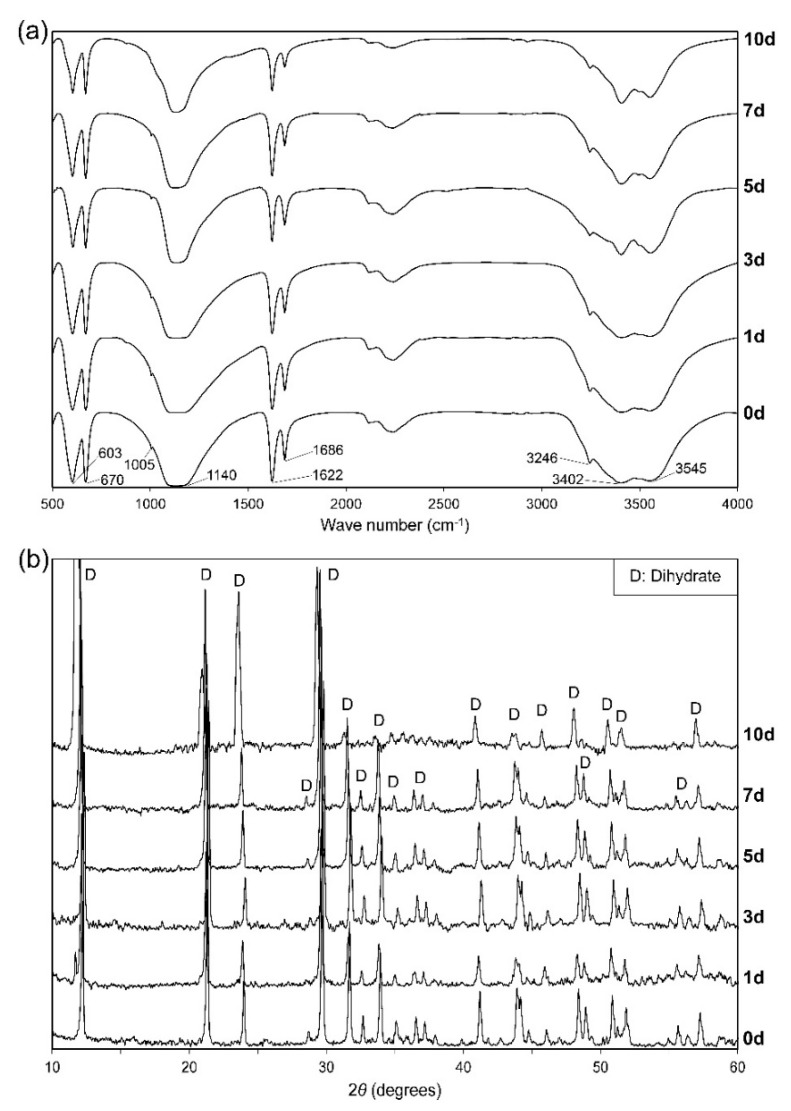
(**a**) FTIR spectra and (**b**) X-ray diffraction patterns of calcium sulfate (CS) cements after 0, 1, 3, 5, 7, and 10 days immersed in Tris–SBF-27 solution.

**Figure 6 materials-13-03099-f006:**
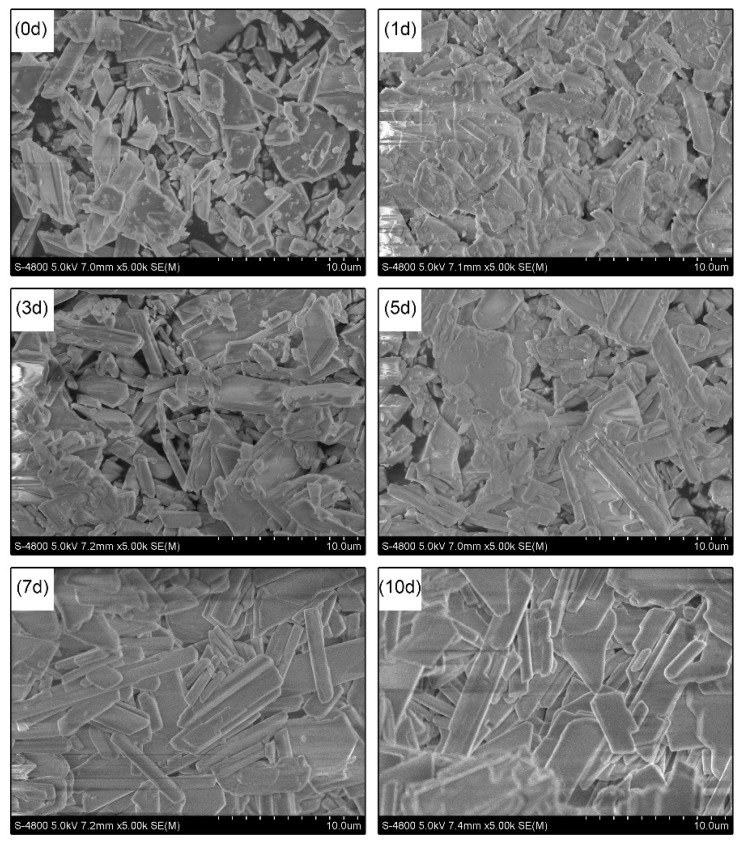
SEM micrographs of CS cements after 0, 1, 3, 5, 7, and 10 days immersed in Tris–SBF-27 solution.

**Figure 7 materials-13-03099-f007:**
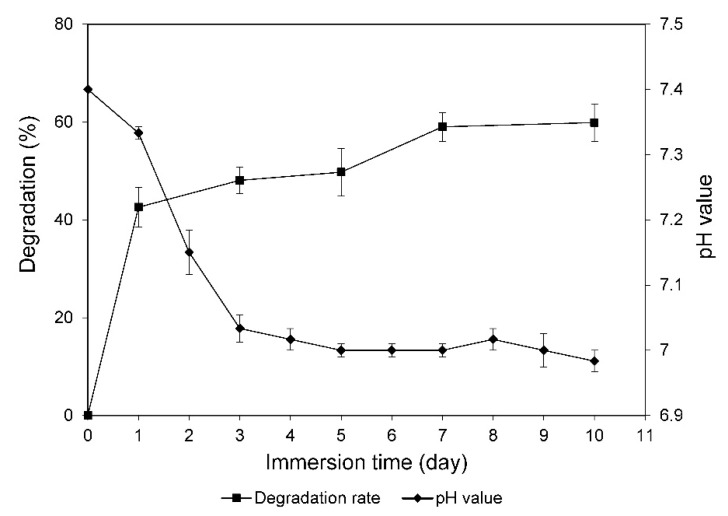
Degradation rate and pH value of CS cements after 0, 1, 3, 5, 7, and 10 days immersed in Tris–SBF-27 solution.

**Figure 8 materials-13-03099-f008:**
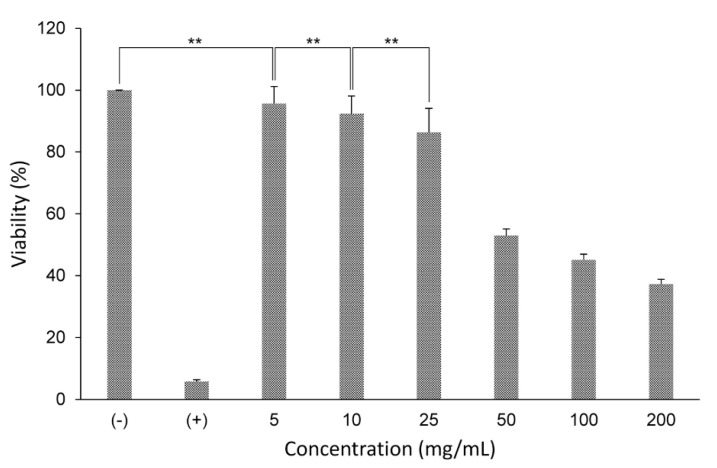
Viability of 3T3 cells after incubation with different CS crystal concentrations: 5, 10, 25, 50, 100, and 200 mg/mL, (−) negative control and (+) positive control. Error bars represent mean ± SD. Statistically significant difference each other (* *p* < 0.05; ** *p* < 0.01)

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
