# Peer review of "A Facile Synthesis Process and Evaluations of α-Calcium Sulfate Hemihydrate for Bone Substitute"

_materials, 2020, doi:10.3390/ma13143099_

Round 1

Reviewer 1 Report

Reviewer report for the article “A Facile Synthesis Process and Evaluations of 3 α-Calcium Sulfate Hemihydrate for Bone Substitute.”

The manuscript focus on α-Calcium Sulfate Hemihydrate synthesized by a proposed simple synthesis process by the autoclave method. The article is interesting and with reliable content, which may contribute to our literature. However, the reviewer has some questions due to some points in content with ambiguous, which need clarification. Therefore, this article may be published after this vague argument becomes clear.

In abstract

  1. the sentence “Despite of those advantages, the synthesis of surgical grade products without the use of any additives, which could cause adverse immune reactions, has not been clearly mentioned yet.” is not clear. Do you mean additives cause adverse immune reactions or the synthesis of surgical grade products cause adverse immune reactions? Please consider to clarity this.
  2. Similarly, in sentence ”The lactate dehydrogenase (LDH) assay was performed showing good biocompatibility of the material, however, its potential for cytotoxicity was also observed in NIH 3T3 cells.” Consider adding a common between “performed” and “showing”.

In introduction

  1. The first sentence show, “Bone regeneration, or fracture healing, is a fascinating process in which bone is able to self-regenerate and reach its initial function without leaving any types of scar tissue [Error! Reference source not found.].” which may indicate software for reference work cannot work correctly. Please check and revised it.
  2. the author mentioned that “The production of α-HH by autoclaving has long been known, however, the output of high purity, especially when not adding any additives in processing, has hardly been studied.” This description is weak in the importance and novelty of this article. Please consider to include some content to describe the autoclave method has been used only at the industrial level. This may confirm the important of this article since they can synthesis calcium–sulphate alpha–hemihydrate for clinical application with an traditional method employed in the industry before. You may check the paper published by Aleksandra Kostić-Pulek, Slobodanka Marinković, Vesna Logar, Rudolf Tomanec, Svetlana Popov in Ceramics − Silikáty 44 (3) 104-108 (2000) Production Of Calcium Sulphate Alpha-Hemihydrate From Citrogypsum In Unheated Sulphuric Acid Solution “as following

“Calcium–sulphate alpha–hemihydrate (α–CaSO4⋅0.5 H2O) can be produced by heating natural or synthetic calcium–sulphate dihydrate (CaSO4⋅2 H2O) in an autoclave to temperature over 100 °C (autoclave method). This method has been used only at the industrial level. The first industries for the autoclave production of alpha–hemihydrate by this method were built in Germany (Guilini) in 1962 and Japan (Nitto) in 1973 [1 – 4]”

References

    1. Zurs A.: J.Amer.Ceram.Soc. 74, 1117 (1991).
    2. Combe E., Smith D.: J.Appl.Chem. 18, 307, (1968).
    3. Combe E., Smith D.: J.Appl.Chem.Biotechnol. 18, 283, (1971).
    4. Singh M., Rai M.: J.Chem.Tech.Biotechnol. 43, 1, (1988).

  1. In the last sentence of the introduction “Cytotoxicity assay was used to investigate the potential toxic effects of α-HH, particularly in the form of calcium sulfate particles, on patients” this sentence may lead to misinterpretations. Do you do cytotoxicity assay in patients?

Do you want to describe the use of particle form of calcium sulfate as clinically used on patients for your cytotoxicity assay?

You may consider changing the sentence as “Cytotoxicity assay was used to investigate the potentially toxic effects of α-HH, particularly in the form of calcium sulfate particles as used clinically.”

In Figure

Figure 6 (7d) (10d) there are some shadow in the SEM picture, could you replace with better ones

Author Response

Dear Reviewer,

Thank you for taking the time to review our manuscript. Your comments is very helpful to improve the quality of this paper. The revisions that have been made included:

In Abstract:

Point 1: The sentence “Despite of those advantages, the synthesis of surgical grade products without the use of any additives, which could cause adverse immune reactions, has not been clearly mentioned yet.” is not clear.

Response 1: Line 20-21, I cut off the additional information: "which could cause adverse immune reactions", so it only mentioned in Introduction.

Point 2: Similarly, in sentence ”The lactate dehydrogenase (LDH) assay was performed showing good biocompatibility of the material, however, its potential for cytotoxicity was also observed in NIH 3T3 cells.”

Response 2: Changed into "The lactate dehydrogenase (LDH) assay showed good biocompatibility of the material, however, its potential for cytotoxicity was also observed in NIH 3T3 cells"

In Introduction:

Point 1: [Error! Reference source not found.]

Response 1: Line 35, revised as requested.

Point 2: "The production of α-HH by autoclaving has long been known, however, the output of high purity, especially when not adding any additives in processing, has hardly been studied.” This description is weak in the importance and novelty of this article.

Response 2: Line 63-64, mentioned autoclave as the only method in industry use DH as raw material. Line 69-71, mentioned the long history of autoclave method but the output of high purity for medical application, especially when not adding any additives in processing, has hardly been studied. In conclusion, we want to emphasize the simple operation and the elimination of additives because there are also some studies that use additives in autoclave method. Combining these two, we want to reduce expenses and associated clinical risks for bone treatments.

Point 3: In the last sentence of the introduction “Cytotoxicity assay was used to investigate the potential toxic effects of α-HH, particularly in the form of calcium sulfate particles, on patients” this sentence may lead to misinterpretations. Do you do cytotoxicity assay in patients?

Response 3: Line 76-77, revised as "Cytotoxicity assay was used to investigate the potential toxic effects of α-HH, particularly in the form of calcium sulfate particles, which can exist clinically in patients". The usage form as cements, which can release particles mentioned in line 317-323.

In Figure:

Figure 6 (7d) (10d) there are some shadow in the SEM picture, could you replace with better ones.

Response: We do have other pictures but at different magnifications. Their quality is not much higher than those two used in the paper. So, if possible, I would like to keep them.

Reviewer 2 Report

While the manuscript is generally will written, i would like to make a few comments 

Comment 1

The authors write (Many in vitro and in vivo studies revealed that CS has always failed to form chemical bonds with bone tissue). In vitro studies can not show this kind of bonding with bone tissue

Comment 2

In the page lines 235-244 authors provides (facts ? are they present in the literature?) on caclium sulfate withot giving appropriate citation. ref 4 is not completely relevent to the all informations given i the text. Relevent och specific citations should be given for these information (used as an inorganic binder to fill defects, acts as a barrier, provides sufficient space for osteoconduction, neovascularization and osteogenesis owning to its porous structure). Lack of citation applies also on the text in the lines 240-244

Comment 3

The authors write in the conclusion  (The cytotoxicity results indicated that the synthesized powder did not influence proliferation behaviors of NIH-3T3). Does LDH test provides information on viability or cell proliferation?

Line 337 Does the authors mean (the instability of CS cements) its degradability ? if it so, maybe better at write degradation instead. 

Author Response

Dear Reviewer,

Thank you for taking the time to review our manuscript. Your comments is very helpful to improve the quality of this paper. The revisions that have been made included:

Point 1: The authors write (Many in vitro and in vivo studies revealed that CS has always failed to form chemical bonds with bone tissue). In vitro studies can not show this kind of bonding with bone tissue.

Response 1: Line 225-226, revised as "Many in vitro and in vivo studies have indirectly or directly revealed that CS has always failed to form bonds with bone tissue". In vitro can reveal indirectly through apatite layers as mentioned in line 226-229.

Point 2: In the page lines 235-244 authors provides (facts ? are they present in the literature?) on caclium sulfate withot giving appropriate citation. ref 4 is not completely relevent to the all informations given i the text. Relevent och specific citations should be given for these information (used as an inorganic binder to fill defects, acts as a barrier, provides sufficient space for osteoconduction, neovascularization and osteogenesis owning to its porous structure). Lack of citation applies also on the text in the lines 240-244.

Response 2: Line 236-246, revised the citations. Line 236-239, discussions drawn from the experiment. Line 239-246, some additional information that the results can effect.

Point 3:

The authors write in the conclusion  (The cytotoxicity results indicated that the synthesized powder did not influence proliferation behaviors of NIH-3T3). Does LDH test provides information on viability or cell proliferation?

Line 337 Does the authors mean (the instability of CS cements) its degradability? if it so, maybe better at write degradation instead.

Response 3:

Line 335, revised as "...synthesized powder did not influence the viability of NIH-3T3 cells..."

Line 338-339, revised as "the instability of CS cements caused by the incomplete setting process leads to the strong release of its particles which might be harmful when in direct contact with host tissues in the body". The incomplete setting process cause the instability in CS therefore the particles strongly release, which should be more cautious than the usual degradation, also mentioned in line 319-323.

Reviewer 3 Report

This article reports the synthesis of calcium hemihydrate for bone substitute applications.

Abstract should be precise. Avoid detailed background information in abstract. Define the problem clearly.

In abstract, what is non-bioactivity?

In check the sentence, "The LDH assay.." Please revise.

In introduction, clearly define the advantage of this method over the conventional methods reported in the literature.

Line 83 and 84, Not sure if they need 55 degree C for 16 hrs to remove acetone from 5 g material.

Line 123-125 may be removed. Background information not required in methods.

Give more details on UV sterilization of the synthesized materials.

The authors could check the degradability of the material in body fluids or corrosion studies using electrochemical techniques. 

Author Response

Dear Reviewer,

Thank you for taking the time to review our manuscript. Your comments is very helpful to improve the quality of this paper. The revisions that have been made included:

Point 1: Abstract should be precise. Avoid detailed background information in abstract. Define the problem clearly.

Response 1: Thank you. I made a small revision

Point 1: In abstract, what is non-bioactivity?

Response 1: Line 25,revised as "no bioactivity"

Point 2: In check the sentence, "The LDH assay.." Please revise.

Response 2: Line 25-26, changed into "The lactate dehydrogenase (LDH) assay showed good biocompatibility of the material..."

Point 3: In introduction, clearly define the advantage of this method over the conventional methods reported in the literature.

Response 3: I changed some words in lines 61-62 and 67-69 to emphasize the novelty.

Line 61-62, mentioned autoclave as the only method in industry use DH as raw material. Line 67-69, mentioned the long history of autoclave method but the output of high purity for medical application, especially when not adding any additives in processing, has hardly been studied. In conclusion, we want to emphasize the simple operation and the elimination of additives because there are also some studies that use additives in autoclave method. Combining these two, we want to reduce expenses and associated clinical risks for bone treatments.

Point 4: Line 83 and 84, Not sure if they need 55 degree C for 16 hrs to remove acetone from 5 g material.

Response 4: May be less time is required. So, futher studies need to be conducted to save costs on this step.

Point 5: Line 123-125 may be removed. Background information not required in methods.

Response 5: revised as requested. I moved the text to Line 290-292.  

Point 6: Give more details on UV sterilization of the synthesized materials.

Response 6: Line 129, revised as: "...sterilized with ultraviolet radiation for 2 hours."

Point 7: The authors could check the degradability of the material in body fluids or corrosion studies using electrochemical techniques.

Response 7: The electrochemical technique is more suitable for inert materials. In this case, the SBF solution will be more acceptable as it can check the bioactivity of bone materials. Also, the SBF are more mimic the body to have a better imagination of how bone material will degrade.

Reviewer 4 Report

Dear Authors,

these are my comments about the article:

1) Line 36: ''Error!:Reference source not found''. Please correct that.

2) Line 153. You wrote ''Turkey's test''. You meant ''Tukey's test''. Please correct that.

3) Line 192: You wrote two times ''27'' in reference bracket. Please correct that.

4) Figure 2. : You wrote ''degees'' instead ''degrees''. Please correct that.

Author Response

Dear Reviewer,

Thank you for taking the time to review our manuscript. Your comments is very helpful to improve the quality of this paper. The revisions that have been made included:

Point 1: [Error! Reference source not found.]

Response 1: Line 35, revised as requested.

Point 2: Line 153. You wrote ''Turkey's test''. You meant ''Tukey's test''.

Response 2: Line 154, revised as requested.

Point 3: Line 192: You wrote two times ''27'' in reference bracket.

Response 3: Line 193, revised as requested.

Point 4: Figure 2. : You wrote ''degees'' instead ''degrees''.

Response 4: revised as requested.

Reviewer 5 Report

The article entitled ‘A Facile Synthesis Process and Evaluations of α-2 Calcium Sulfate Hemihydrate for Bone Substitute’ is well written and organized. The quality of experimental work is good as well as the scientific interpretations. I recommend the authors to review the text to improve the quality of the actual manuscript before it is accepted

  1. The proposed literature is adequate but in reviewer’s opinion the authors should clearly emphasise the novelty of their study and how they delineate themselves from previous works. The processing method looks quite similar like in the reference 13.
  2. Does the lack of band at 1680 cm–1 in Fig 1 indicate the absence of loosely held water molecules in CS and therefore indicates the hemi- and anhydrous form of calcium sulphate? In the FTIR interpretation there is lack of assignment to specific types of vibrations (stretching/symmetrical/out-of-plane/bending ect.)
  3. The Raman spectra and NMR spectroscopic investigations can help to show structural differences between α- and β-hemihydrates.
  4. Based on XRD pattern (Fig. 2), the authors could estimate the percentage of α- and β-hemihydrates using for example Retvield method and check if there is correlation between the TGA results.
  5. Line 190: there is small error in the references.
  6. The lack of bioactivity is also the concequence of decrease of pH value.
  7. The low cells viability for 50, 100 and 200 mg/mL can be also caused by too low pH values. Is it possible for the authors to show what was the pH of each sample solution used in LDH test?
  8. In Fig 8, the author could show the sinificanat differences between 10 and 25 mg/mL to compare with negative control, not only between specific samples.

Author Response

Dear Reviewer,

Thank you for taking the time to review our manuscript. Your comments is very helpful to improve the quality of this paper. The revisions that have been made included:

Point 1: The proposed literature is adequate but in reviewer’s opinion the authors should clearly emphasise the novelty of their study and how they delineate themselves from previous works. The processing method looks quite similar like in the reference 13.

Response 1: I changed some words in lines 61-62 and 67-69 to emphasize the novelty.

Line 61-62, mentioned autoclave as the only method in industry use DH as raw material. Line 67-69, mentioned the long history of autoclave method but the output of high purity for medical application, especially when not adding any additives in processing, has hardly been studied. In conclusion, we want to emphasize the simple operation and the elimination of additives because there are also some studies that use additives in autoclave method. Combining these two, we want to reduce expenses and associated clinical risks for bone treatments.

About the reference 13, E. C. Combe and D. C. Smith reported on well-formed α-HH adding sodium succinate with 99.2% purity. So, they actually added an additive in processing.

Point 2: Does the lack of band at 1680 cm–1 in Fig 1 indicate the absence of loosely held water molecules in CS and therefore indicates the hemi- and anhydrous form of calcium sulfate? In the FTIR interpretation there is lack of assignment to specific types of vibrations (stretching/symmetrical/out-of-plane/bending ect.)

Response 2:

Line 157 mentioned as "1620 cm-1 for bending vibrations of a single type of water in HH molecular structure". Line 207 mentioned as "Two bending vibrations at 1686 and 1622 cm-1 indicated the presence of two types of water molecules in DH crystals". So, the lack of band at 1680 cm–1 in Fig 1 indicated the absence of one water molecule. (CaSO4.2H2O and 2CaSO4.H2O)

Line 156-158, 207-210, revised as requested for the lack of assignment to specific types of vibrations.

Point 3: The Raman spectra and NMR spectroscopic investigations can help to show structural differences between α- and β-hemihydrates.

Response 3: Thank you. Yes, they can. However, due to lack of conditions and technical equipment, we could not afford it. So, I utilized the XRD result instead of having to do another test.

Point 4: Based on XRD pattern (Fig. 2), the authors could estimate the percentage of α- and β-hemihydrates using for example Rietveld method and check if there is correlation between the TGA results.

Response 4: Rietveld method is excellent potential way to find the purity. However, compared with TGA, it is still not standardized at the moment for hydrated cement phases. There have been studies on the use of this method and there is still a certain deviation from the results from TGA (DOI: 10.1154/1.1649328). Therefore, I think the TGA result is sufficient for the main purpose of our study.

Point 5: Line 190: there is small error in the references.

Response 5: Line 188, revised as requested.

Point 6: The lack of bioactivity is also the concequence of decrease of pH value.

Response 6: Line 96-98, the lack of bioactivity is due to the lack of apatite-forming ability. Apatite support bone ingrowth and osseointegration in human body.

Point 7: The low cells viability for 50, 100 and 200 mg/mL can be also caused by too low pH values. Is it possible for the authors to show what was the pH of each sample solution used in LDH test?

Response 7: Your comment is a good factor for us to continue researching. As mentioned in Line 261, the decrease in pH are basically came from the Ca ions. Therefore, based on the literatures, we assumed it as the main reason. Also, I think more detailed studies should be conducted to give exact reasons. For example, perhaps a material that does not contain calcium ions also lowers the pH but does not affect the cell viability.

Point 8: In Fig 8, the author could show the sinificanat differences between 10 and 25 mg/mL to compare with negative control, not only between specific samples.

Response 8: Line 301-304, I added some information as "No significant difference in cell viability was observed between the 5 mg/mL-treated group compared to negative controls and the 10 mg/mL-treated group as well as between the 10 mg/mL-treated group compared to the 25 mg/mL-treated group (p < 0.01)". Therefore, compared with the negative control, only the concentration of 5 mg/mL exhibited no significant different.

Round 2

Reviewer 3 Report

The authors have not made significant efforts to improve the article.

Therefore, I recommend revising this article extensively.

Author Response

Dear Reviewer,

Thank you for taking the time to review our manuscript. Your comments is very helpful to improve the quality of this paper. We are trying our best to improve this manuscript. The revisions that have been made included:

Point 1: Abstract should be precise. Avoid detailed background information in abstract. Define the problem clearly.

Response 1: In the abstract, I included brief background, method, results and conclusions. Also, I removed the excess information and defined the problem as "the synthesis of surgical grade products without the use of any additives, which could cause adverse immune reactions, has not been clearly mentioned yet".

Point 2+3: In abstract, what is non-bioactivity? In check the sentence, "The LDH assay.." Please revise.

Response 2+3: I suppose you're referring to the grammar problem so I've modified as follows: Line 24 "no bioactivity" and Line 26 "The lactate dehydrogenase (LDH) assay showed good biocompatibility of the material..."

Point 3: In introduction, clearly define the advantage of this method over the conventional methods reported in the literature.

Response 3: I revised in line 76-79 to emphasize the novelty.

"The first commercial processes for α-HH production were used in building construction in Germany, Japan and America [16,22]. As a traditional method used in industry, the production of α-HH by autoclaving has long been known [13], however, the output of high purity for medical application, especially when not adding any additives in processing, has hardly been studied".  So the synthesis calcium–sulphate alpha–hemihydrate for clinical application with an traditional method employed in the industry before is the important of this article.

Point 4: Line 83 and 84, Not sure if they need 55 degree C for 16 hrs to remove acetone from 5 g material.

Response 4: Perhaps the observation both variables as time and temperature simultaneously is necessary to obtain optimal conditions at this step. So, in the next study we may deploy it.

Point 5: Line 123-125 may be removed. Background information not required in methods.

Response 5: revised as requested.

Point 6: Give more details on UV sterilization of the synthesized materials.

Response 6: Line 138, The additional information is execution time "...for 2 hours"

Point 7: The authors could check the degradability of the material in body fluids or corrosion studies using electrochemical techniques.

Response 7: The electrochemical technique is more suitable for inert materials. In this case, the SBF solution will be more acceptable as it can check the bioactivity of bone materials. Also, the SBF are more mimic the body to have a better imagination of how bone material will degrade.

Reviewer 5 Report

The manuscript can be published in the present form.

Author Response

Dear Reviewer,

Thank you for taking the time to review and accept our manuscript in the present form.